# Positive Effect of Cognitive Training in Older Adults with Different *APOE* Genotypes and COVID-19 History: A 1-Year Follow-Up Cohort Study

**DOI:** 10.3390/diagnostics12102312

**Published:** 2022-09-25

**Authors:** Yana Zorkina, Timur Syunyakov, Olga Abramova, Alisa Andryushchenko, Denis Andreuyk, Evgeniya Abbazova, Dmitry Goncharov, Alisa Rakova, Nika Andriushchenko, Dmitry Gryadunov, Anna Ikonnikova, Elena Fedoseeva, Marina Emelyanova, Kristina Soloveva, Konstantin Pavlov, Olga Karpenko, Victor Savilov, Marat Kurmishev, Olga Gurina, Vladimir Chekhonin, Georgy Kostyuk, Anna Morozova

**Affiliations:** 1Mental-Health Clinic No. 1 Named after N.A. Alekseev, Zagorodnoe Highway 2, 115191 Moscow, Russia; 2Department of Basic and Applied Neurobiology, V. Serbsky Federal Medical Research Centre of Psychiatry and Narcology, Kropotkinsky per. 23, 119034 Moscow, Russia; 3International Centre for Education and Research in Neuropsychiatry (ICERN), Samara State Medical University, 443016 Samara, Russia; 4Faculty of Economics, M.V. Lomonosov Moscow State University, 119991 Moscow, Russia; 5N.F. Gamaleya Federal Research Center for Epidemiology and Microbiology, Ministry of Health of Russia, Gamaleya st. 18, 123098 Moscow, Russia; 6Department of Biology, Shenzhen MSU-BIT University, Ruyi Rd. 299, Shenzhen 518172, China; 7Center for Precision Genome Editing and Genetic Technologies for Biomedicine, Engelhardt Institute of Molecular Biology, Russian Academy of Sciences, 119991 Moscow, Russia; 8Federal State Budgetary Educational Institution of Higher Education, Moscow State University of Food Production, Volokolamskoye Highway 11, 125080 Moscow, Russia

**Keywords:** MCI, COVID-19, SARS-CoV-2, *APOE*, rs429358, rs7412, MMSE, HADS, dementia progression, long-term study

## Abstract

(1) Background: Older people suffer from cognitive decline; several risk factors contribute to greater cognitive decline. We used acquired (COVID-19 infection) and non-modifiable (presence of *APOE* rs429358 and rs7412 polymorphisms) factors to study the progression of subjective cognitive impairment while observing patients for one year. Cognitive training was used as a protective factor. (2) Methods: Two groups of subjects over the age of 65 participated in the study: group with subjective cognitive decline receiving cognitive training and individuals who did not complain of cognitive decline without receiving cognitive training (comparison group). On the first visit, the concentration of antibodies to COVID-19 and *APOE* genotype was measured. At the first and last point (1 year later) the Mini-Mental State Examination scale and the Hospital Anxiety and Depression Scale were performed. (3) Results: COVID-19 infection did not affect cognitive function. A significant role of cognitive training in improving cognitive functions was revealed. Older adults with *APOE-ε4* genotype showed no positive effect of cognitive training. (4) Conclusions: Future research should focus on cognitive dysfunction after COVID-19 in long-term follow-up. Attention to the factors discussed in our article, but not limited to them, are useful for a personalized approach to maintaining the cognitive health of older adults.

## 1. Introduction

The aging of the population in developed countries around the world is currently taking place. Older people often suffer from cognitive decline: difficulties with memory, learning and later inability to care for themselves. Cognitive impairments seen in aging range from minor cognitive impairment to dementia [1].

There are several risk factors for developing more severe cognitive impairment, age, gender, education and biological risk factors.

In some cases, the degree of cognitive impairment in aging is classified as mild cognitive impairment (MCI). The term MCI represents a concept that reflect prodromal states of dementias of various underlying etiology [2]. This concept emerged as a reaction to the irreversibility of dementia developing with ageing due to neuro-progressive brain disease in order to capture those diagnostically dubious cases that should be followed and treated. According to studies, the prevalence of MCI in adults over 60 years of age is approximately 6.7% to 25.2% [3,4]. People with MCI are thought to be 3–5 times more likely to progress to any form of dementia than people with normal aging, with an annual progression rate of 12% in the general population and up to 20% in high-risk groups [4]. Two types of MCI, amnestic and non-amnestic, were differentiated based on the prevalence of symptoms with amnestic type along with specific biomarkers highly suggestive of Alzheimer’s disease (AD) [5] with overall risk of conversion into severe dementia of approximately 30% [6].

The main idea in identifying these intermediate stages of neuro-progression to dementia was the need for a timely identification of underlying disorder when a risk factor correction or pathogenetic intervention could be applied to prevent negative scenarios [3]. Another goal of studying MCI is to try to differentiate disease-specific subtypes of MCI that can lead to specific forms of dementia. This could help to develop effective targeted/stratified treatment strategies or prevention measures aimed at risk factors correction. Among risk factors of progression to dementia, some are inherent and can’t be modified while others are acquired. The latter, or their implications, could potentially be modified by prevention or treatment strategies. Non-modifiable risk factors include primarily age and genetics. To date, ε-polymorphism of the apolipoprotein E *(APOE)* gene remains the strongest genetic factor affecting the risk of AD after numerous large-scale genomic association studies and genomic association meta-analysis [7]. The *APOE ε4* allele is found in 20–25% of AD patients, increasing the risk of the disease three-fold in heterozygous carriers and 15-fold in homozygous carriers [8]. The *APOE ε4* allele is also associated with more severe tau pathology with higher Aβ plaque burden on PET studies and postmortem samples [7]. The *APOE ε2* allele, on the contrary, is a significant protective factor [7].

There are studies supporting evidence that the presence of *APOE*
*ε4* allele increases the risk of MCI symptoms rapidly worsening into dementia [9]. Interactions between *APOE ε4* and MCI led to more severe pathological symptoms [10].

In our study we consider COVID-19 infection as an acquired risk factor.

The possible cognitive effects of coronavirus disease are explained by the capacity of the virus to penetrate the blood–brain barrier, interacting with endotheliocytes, nerve cells and glia and causing neuroinflammatory reactions [11,12]. Over time, glial changes lead to cellular stress, ultrastructural changes in organelles (e.g., mitochondria) and functional abnormalities that become increasingly pronounced in adulthood and aging [12].

Coronavirus infection impairs neuroimmune axis function, which may be involved in cognitive impairment in mental illness and increases levels of IL-1β and IL-6 in the brain, both of which tend to suppress synaptic plasticity, affecting learning and memory [13].

The most frequently disclosed psychiatric symptoms in coronavirus are low mood, mood swings, hopelessness, heightened anxiety, sleep/wake cycle dysregulation and neurocognitive disturbances including brain fog, difficulties with memory, concentration and executive function [14]. It is also important to mention cognitive impairment following resolution of acute COVID-19 [15]. About two-thirds of patients hospitalized with COVID-19 demonstrated clinically significant cognitive impairment even 4 months after discharge [16]. There was also an increased incidence of dementia, intra-cranial hemorrhage and ischemic stroke within the first 6 months after COVID-19 diagnosis in a retrospective cohort study and those risks were proportional to infection severity [17]. A meta-analysis of subjects infected with SARS-CoV-2 showed cognitive decline regardless of stage or age up to 1 year of follow-up [18]. Patients with pre-existing mental illness or cognitive impairment, as well as older adults, are most susceptible to the negative effects of coronavirus on the nervous system. A retrospective study showed a significant increase in diagnoses of neuropsychiatric disorders, including depression, anxiety, insomnia and psychosis during lockdown. Moreover, patients with neuropsychiatric disorders are particularly affected by the psychological effects of social distancing, isolation and quarantine measures [19]. A study by Cagnin et al. [20] showed impaired behavioral and psychological symptoms in approximately 60% of patients with dementia within 1 month of quarantine in 2020.

In turn, as has been shown, the *APOE* gene polymorphism can affect susceptibility to SARS-CoV-2 infection, COVID-19 severity and mortality [21,22,23]. The *APOE* ε4 allele was a risk factor for severe COVID-19 and post-COVID mental fatigue [21,23], whereas protective effect of *ε2* allele against SARS-CoV-2 infection was shown [22]. In the brain autopsy material of COVID-19 patients, perivascular microhemorrhages were found to be more common in *APOE ε4* carriers, suggesting that some of these effects may be mediated by increased cerebrovascular damage [21].

It is assumed that the long-term cognitive and neurological effects of COVID-19 infection may also depend on the clinical course of the disease and the *APOE* genotype [24,25], but this needs to be confirmed in prospective studies. For example, it has been hypothesized that COVID-19-induced olfactory dysfunction may cause an increased risk of future neurodegenerative dementia in ApoE4 carriers [24]. Prospective studies of the role of genetic markers in the context of COVID-19 disease and clinical symptoms seem appropriate and may help identify risk groups, as well as elucidate the mechanisms of pathological changes in COVID-19 infection.

One of the main strategies to intervene against acquired risk factors and reduce the progression of MCI into dementia is cognitive training, involving training memory, attention and other mental functions. Conducting various neurocognitive rehabilitation programs has a positive effect on the cognitive functions of patients with MCI. A study by Peng et al. showed significant cognitive improvement 6 months after starting training compared to the control group [26]. Several meta-analyses are also focused on this issue. The positive effect of multicomponent training [27,28,29] and exercise [30,31] was shown.

According to the new paradigm of positive gerontology, the emphasis is not on dis-ease as an obligatory link to aging, but on the ability to use mental resources in the third and fourth age. The development of modern approaches to rehabilitation of cognitive deficits uses predictive models with multimodal markers, which is important for medical professionals for diagnosis and identification of therapeutic and rehabilitation targets, as well as for non-medical specialists who work with the problem of cognitive competence decline at the stage of late ontogenesis. Genetic and environmental factors are being studied both in connection with the development of neurodegenerative and cerebrovascular diseases and in connection with the re-organization of cognitive resources. The coronavirus pandemic is a new and serious factor in cognitive decline, the role of which requires a long study.

The aim of our study was to investigate association between COVID-19 infection and *APOE* genotype as factors contributing to cognitive decline in older adults with a follow-up period of 1 year. An additional aim of our work was to investigate the assumption of a positive effect on cognitive function of the rehabilitation program in the presence of other risk factors.

## 2. Materials and Methods

### 2.1. Subjects

The study included individuals observed at the N.A. Alekseev Clinical Hospital No.1 from September 2020 to February 2022. The first point of observation was the end of 2020 (15 September 2020 for the first subject and the first follow-up took place on 7 October 2021).

Two groups of subjects over the age of 65 participated in the study:People who came to the “Memory Clinic” of Mental-health Clinic No. 1, named after N.A. Alekseev of the Moscow Healthcare Department, with subjective cognitive decline. These subjects received cognitive training. The study included individuals with complaints of forgetfulness, lack of attention and concentration (for example, when talking or reading a book), episodic difficulties in finding their way home, feeling if hard to articulate their thoughts, decreased professional and social productivity, impaired motor skills (writing, drawing) and experiencing problems in everyday life (paying bills, shopping).People who came to outpatient clinic No 121 (Moscow) to receive COVID-19 vaccination. This group of subjects did not complain of cognitive decline. No cognitive training was conducted with this group (comparison group).

On the first visit, the concentration of antibodies to COVID-19 and *APOE* genotype was measured. At that observation point, there were no COVID-19 vaccinated individuals among the subjects. At the first and last follow-up point, cognitive tests were performed using the Mini-Mental State Examination (MMSE) scale [32] and clinical symptomatology of COVID-19 was assessed. Depression level was assessed with The Hospital Anxiety and Depression Scale (HADS) [33]. The period between the two points was 1 year.

Subjects were excluded due to any of the following: dementia, psychiatric illness, positive family histories (first-degree relatives) of psychiatric illness; substance abuse; severe somatic diseases.

### 2.2. Neurocognitive Training

As part of the federal project “The Older Generation” and the regional program “Active Longevity”, in 2016 Moscow organized a network of “Memory Clinics”, consisting of clinical and rehabilitation units for day care patients. An interdisciplinary team approach involving psychiatrists, psychologists, medical and social workers was used to implement a comprehensive medical and rehabilitation program.

Fast recovery of individuals with cognitive decline after the medical and rehabilitation program and restoration of all components of the higher mental functions is shown; an adaptation of the program taking into account conditions of coronavirus pandemic did not demand increased duration, nor cardinal change in the form of training. The duration of the rehabilitation cycle is 6 weeks, 96 h of sessions, with a clear principle of progressive complication and repeated practice of structured tasks to improve certain cognitive functions. Rehabilitation was conducted in a group format. Groups of 8–9 persons were formed taking into account age and cognitive similarity. The first module of cognitive training, “psycho-education”, lasts 1 week, then during 2–6 weeks cognitive-therapeutic sessions include neurocognitive training twice a day for 60 min and classes of “adaptive physical training” (in modes on Barthel’s scale: sparing (86–102 points) or sparing (70–85 points)) and psychotherapy. Neurocognitive training is aimed at restoration of visual-spatial gnosis, speech, mnestic, kinesthetic, tactile and somatognostic processes, attention, goal setting, control and the activation of energetic processes of the organism [34]. Psychotherapy includes methods of psychological aid adapted to the rehabilitation program (cognitive-behavioral psychotherapy, cognitive psychodrama).

### 2.3. APOE Genotyping

Genomic DNA samples were obtained from peripheral blood lymphocytes using an automated DNA extraction system (QIAGEN QIAcube, Germany) according to the manufacturer’s recommendations. The *ε2/ε3/ε4* alleles of the *APOE* gene were determined by real-time PCR based on genotyping for the rs429358 and rs7412 markers as described previously [35].

### 2.4. ELISA

IgG antibodies to SARS-CoV-2 in subjects’ serum were determined using SARS-CoV-2-IgG-IFA-BEST and SARS-CoV-2-IgM-IFA-BEST test systems (VECTOR-BEST JSC, Russia), providing positivity rate as a semi-quantitative measurement result.

### 2.5. Statistical Processing

Study population was characterized by means of descriptive statistics providing means and standard deviations (S.D.) for continuous variables and absolute and relative frequencies for categorical variables. Between-group comparisons were performed using One-way ANOVA for continuous variables and Pearson Chi-square test for categorical variables.

Full-factorial repeated-measures ANOVA model was used to assess effects of group, ant-COVID-19 IgG status and *APOE-ε4* polymorphism on pre-post changes in total MMSE score.

To assess how *APOE* polymorphism and COVID-19 affected changes in cognitive scale total score from background to endpoint across groups, we first built a regression model with dependent variable of MMSE total score changes, with group as fixed factor and age and baseline MMSE score as a covariate. As the second step, we separately added to this model a number of *ε2* and *ε4* alleles of the *APOE* as factors and IgG against SARS-CoV-2 as a covariate and evaluated gain in R^2^. Figure 1 describes the statistical analysis models used.

Since depression may cause cognitive dysfunction and COVID-19 is associated with subsequent depression, we used ANCOVA to check ho background depression score on HADS scale as the covariant affected 1-year MMSE changes between groups and used Pearson correlation and one-way ANOVA to check if COVID-19 antibodies (measured quantitatively or qualitatively) influenced the HADS depression score.

## 3. Results

Graphical scheme of research and testing of statistical hypotheses is shown in Figure 1.

Complete data were obtained from 125 people who applied for the cognitive training program and 34 in the comparison group. General characteristics of the studied population are shown in Table 1. Statistically significant differences in age, sex and total baseline MMSE score were revealed and age and MMSE scores were entered into the analyzed models as covariates.

Table 2 shows the distribution of participants in the cognitive training and comparison groups according to *APOE* polymorphisms and presence status and quantitative IgG content for COVID-19. Participants in the groups differed in their COVID-19 antibody content. Figure 1 shows the MMSE total score changes depending on the *APOE* genotype.

Repeated-measures ANOVA (Table 3) revealed significant effect of time x group interaction, without significant effect of any other interaction. Marginal means of MMSE total scores for three-way interactions time × group × *APOE-**ε4* genotype and time × group × anti-COVID-19 IgG status and four-way time × *APOE-**ε4* genotype × anti-COVID-19 IgG status × group interaction are presented in Figure 2. Due to small subsamples size, some contrasts in the models could not be estimated.

Group factor showed significant effect in the changes of MMSE total score from baseline to endpoint (Figure 2A) with significant improvement in cognitive training group (from 26.3 (25.7; 26.8) to 27.5 (26.8; 28.1), *p* < 0.001) and arithmetically but not statistically significant deterioration in the comparison group (from 27.0 (26.0; 28.0) to 26.5 (25.3; 27.6), *p* = 0.312). Status of anti-COVID-19 IgG (Figure 2B) didn’t affect direction of changes of mean total MMSE score changes during 1-year follow-up within cognitive training or comparison groups.

Independently of anti-COVID-19 IgG status, and despite that there was no significant effect of *APOE-**ε4* genotype in wild type allele carriers and heterozygotes in the training group in contrast to comparison group, a general improvement in MMSE total scores from baseline to endpoint was observed (Figure 2C,D). In addition, in all mutant allele carriers (*n* = 5, three from training group and two from comparison group) there was no improvement of MMSE total score during follow up period.

The ANCOVA showed no statistically significant effect of the initial HADS depression score on changes in the MMSE total score (Table 4).

Initial HADS depression score was not associated with IgG antibodies to COVID-19 (r = −0.051, *p* = 0.622) and did not differ between participants with positive and negative test for anti-COVID-19 IgG (Sum of Squares = 23.54, F = 2.29, *p* = 0.13, η_p_^2^ = 0.02).

The results of linear regression of MMSE changes with group as the fixed factor and age and background MMSE scores as covariates were significant (R^2^ = 0.25, *p* < 0.001), indicating that group factor significantly influenced 1-year MMSE total score changes after adjustment in age and background MMSE scores.

Based on regression estimates effect, of group on MMSE score change was largest in the model, followed by MMSE background total score and age (Table 5).

Table 6 shows results of adding of number of *ε2* and ε4 alleles of the *APOE* as factors or anti-SARS-CoV-2 IgG count as a covariate in the model. Adding of *ε4* allele significantly changed initial model (R^2^ change = 0.04, *p* = 0.02), whereas adding of *ε2* allele or anti-SARS-CoV-2 IgG count did not affect initial model results significantly.

## 4. Discussion

We conducted a long-term study of cognitive function changes in older adults over 1 year under the influence of acquired (COVID-19 infection) and non-modifiable (presence of *APOE* rs429358 and rs7412 polymorphisms) factors. We also evaluated the effect of the cognitive training program on MMSE scores, taking these factors into account.

According to the results of our study, the individuals in the cognitive training group statistically significantly differed from the comparison group in changes of MMSE total score. In fact, those subjects, who underwent cognitive training experienced a statistically significant improvement in the MMSE total scale score, in contrast with no statistically significant change but an arithmetical deterioration in the comparison group. This corresponds to the results of other studies that have shown a favorable effect of neurocognitive training on cognitive functioning both in patients with MCI and with subjective cognitive symptoms [34].

It should be stressed that this result was achieved after a 1-year follow-up: that is, 11 months after completion of the 1-month program in the Memory Clinic. Nevertheless, it is not clear whether these favorable changes were a product of continuous improvement, or they occurred soon after the cognitive training program, subsequently remaining at the level achieved or smoothly deteriorating from it.

The factor of having anti-COVID-19 IgG at the start of the study had no significant effect on the 1-year changes in the total MMSE score. However, in this study, we did not test for cases of COVID-19 during the 1-year follow-up period. Despite this, it seems important that there was improvement in the cognitive training group regardless of the presence of anti-COVID-19 antibodies, although participants in this group may also have had COVID-19 during the study period.

The *APOE ε4* genotype carrier factor in our study was not associated with significant differences in the changes in the MMSE total score between groups. At the same time, five *APOE ε4* allele carriers showed no improvement in their MMSE total score regardless of whether they received cognitive training or not.

However, the absence of statistically significant interactions does not mean that the factor has no effect on the results. Therefore, we performed a regression analysis with consistent addition of additional factors to the model.

Cognitive training is a rehabilitation program carried out in our hospital. According to previous studies, carrying out such rehabilitation reduces severity of cognitive manifestations and their progressive deterioration [36]. The goal of our strategy of neurocognitive rehabilitation is to improve the quality of life of older and senile citizens, increase the period of active longevity and healthy life expectancy, restore cognitive deficits and prevent dementia. An interdisciplinary team approach involving psychiatrists, psychologists, medical and social workers was used to implement a comprehensive medical and rehabilitation program. Our training is multi-domain and includes lifestyle changes, as according to meta-analysis [28,29] this type of training is most effective.

Therefore, the results of the neuropsychological study of the cohort of subjects with subjective cognitive decline after completing a course of neurocognitive correction in the conditions of a pandemic coronavirus infection indicate a comprehensive improvement in their cognitive sphere.

An important result of the study is the fact that a significant improvement in the neuropsychological status of the individuals was present irrespective of their age. Subjects with lower grades of cognitive functions at the initial examination showed a more pronounced improvement of indicators after completing a course of neurocognitive correction.

A great deal of data suggests cognitive decline after the coronavirus infection. Some authors believe that the key problem with COVID-19 infection is a long-term neurological impairment [16]. Even with mild COVID-19 six months after symptom onset, the authors found a reduction of ≥4 points on the MoCA scale [37]. Another study found 80% of cognitive impairment using the MMSE, MoCA and Hamilton Rating Scale for Depression and Functional Independence Measure [38]. According to Rass et al. [39] cognitive impairment was observed in 23% of COVID-19 individuals (in severe COVID-19 29%, moderate 30% and mild 3%).

Older adults are at high risk of developing severe forms of COVID-19 because of factors associated with aging and a higher prevalence of comorbid medical conditions and therefore they are more vulnerable to possible long-term neuropsychiatric and cognitive impairment [40]. Chronic conditions such as dementia are of particular concern, not only because they are associated with higher rates of hospitalization and mortality, but also because COVID-19 further exacerbates the vulnerability of people with cognitive impairment. Miskowiak et al., 2021, believe that systematic cognitive screening of patients after recovery from severe COVID-19 disease and implementation of targeted treatment for patients with persistent cognitive impairment is necessary [16]. A study in China found that SARS-CoV-2 infection was associated with an increased risk of cognitive decline in older adults 6 months after the recovery [41].

There is a significant association between Alzheimer’s disease, increased risk of COVID-19 infection and odds of mortality. However, MCI is not a risk factor for SARS-CoV-2 infection [42]. A study conducted in South Korea evaluated the number of newly diagnosed cases of dementia and the worsening of comorbid psychiatric symptoms in subjects with dementia who tested positive for COVID-19. There were three groups of patients: those who tested negative for COVID-19, those who tested positive and a control group who did not take the test. Based on the results of the study, the authors showed that there was no difference in the incidence of dementia between the positive and negative groups, although the incidence of MCI was higher in the positive group than in the negative group. People who already had dementia were more likely to develop comorbid mental disorders in the negative group than in the control group, but less than in the positive group [43]. A meta-analysis of Soysal et al. [44] suggests that neuropsychiatric symptomatology is present and worsens with COVID-19 isolation and pandemic in patients with dementia. Of 21 studies analyzed including 7139 subjects (60.0% women, mean age 75.6 ± 7.9 years, 4.0% MCI) with dementia, five showed no change in neuropsychiatric symptomatology, but all others showed an increase in at least one symptom or in the Neuropsychiatric Inventory score. The most common symptoms were depression, anxiety, agitation, irritability and apathy. The mean follow-up time was 5.9 ± 1.5 weeks, which was not enough to consider long-term effects. In addition, according to the authors, all studies had a high risk of bias [44]. In another study, patients with MCI and AD without a COVID-19 diagnosis did not show significant changes in mood and behavior during quarantine, as the decline in relevant measures from 2019 to 2020 was not different from the decline from 2018 to 2019 [45].

Cognitive function assessment using a regression model and the MMSE score showed no associations between the presence of SARS-CoV-2 IgG antibodies and worsening of cognitive symptoms one year after follow-up. We found no statistically significant differences between the COVID-19 infection and the HADS score.

The systematic review found a high incidence of cognitive impairment after SARS-CoV-2 infection [46]. Most of the selected studies evaluated previously hospitalized patients or even individuals during hospitalization, which could suggest that cognitive impairment might be caused by the acute phase of the disease, hypoxia during infection, etc. In addition, the authors emphasize that most of the works did not include previous cognitive assessment of patients and control groups.

There are few studies focused on the effect of coronavirus on symptomatology and cognitive deterioration in older adults, individuals with MCI, dementia and AD, especially with a long follow-up period.

In the Tsatali [45] study of 407 patients with MCI and AD with follow-up in 2018, 2019 and 2020, no increased cognitive or functional decline was observed during the isolation period with the exception of tests measuring verbal memory, learning and phonemic fluency and daily functioning. However, the authors suggested that the natural progression of MCI and AD was the main cause of the participants’ impairment in the above-mentioned tests [45].

Thus, no long-term effect of SARS-CoV-2 infection at the follow-up period of 1 year in our study is quite consistent with other works. The presence of cognitive impairment shown by other authors was observed at shorter periods, which might be due to acute infection, inflammation and hypoxia phenomena that pass after some time, causing no long-term cognitive effects. Subjects in our study underwent cognitive rehabilitation in the Memory Clinic, which could also contribute to neutralization of the neuropsychiatric effects of COVID-19 and more effective rehabilitation.

A meta-analysis of 5709 individuals showed that the *APOE ε4* allele was also associated with an increased risk of MCI, while the *APOE* ε2/ε3 genotype offered little protection against MCI [47]. A meta-analysis involving 58,000 subjects showed that men and women with the *APOE* ε3/ε4 genotype had almost the same chance of developing AD between the ages of 55 and 85, but women had an increased risk of developing AD at a younger age. *APOE ε4* homozygotes also had an increased risk compared with *ε4* heterozygotes for MCI and for the transition from MCI to AD [48].

There is no evidence that *APOE ε4* alleles are associated with the presence of depression, anxiety, apathy, agitation, irritability, or sleep disturbances in people with cognitive impairment [49]. However, the presence of these symptoms is often observed in subjects with dementia and may be a prodromal stage of the disease. Very few studies focus on the impact of the convergence of neuropsychiatric symptoms and *APOE ε4* allele on the conversion to dementia in individuals with MCI. Valero et al. [38] followed 1512 subjects (age 60 years and older) with MCI for 2 years. Additive interactions were obtained for depression, apathy, anxiety, agitation, appetite, or irritability and positive *ε4* carrier status, which significantly increased the risk ratio for dementia. A combination of behavioral status and genetic trait could be considered to identify subjects with MCI most likely to progress to dementia [50].

Most studies are aimed at finding associations with the diagnosis of MCI or AD, with a confirmed *APOE* genotype. However, the degree of cognitive impairment progression in individuals with different genotypes, as well as studies of impairment in specific cognitive domains, are of greater practical interest.

The Whitehair [51] study over a 36-month period revealed a more rapid decline in performance on all cognitive and functional tests except Number Cancellation and Maze tracing in subjects with the *APOE ε4* genotype. The greatest decline was seen in global examination of cognition and function including the Clinical Diagnostic Rating scale, followed by the MMSE, Global Deterioration scale and the Alzheimer’s Disease Assessment Scale-Cognitive Subscale.

In another study, scores on verbal memory tests were shown to be lower at an average of 33 months in *ε4* carriers compared to noncarriers, while no differences were found in language, spatial skills and executive functions [52].

In a study that lasted 4 years and included 39 individuals (25 converters, 14 nonconverters), MCI subjects who developed dementia, had lower scores on tests of long-term visual memory and semantic fluency. MCI subjects who developed dementia were more likely to have at least one copy of the *APOE ε4* allele [53].

It is important to note that in our study we did not aim to test correlations between cognitive decline and the *APOE* genotype, as this is well known from the literature. We based our analysis on age-appropriate cognitive changes in subjects and the available polymorphism. In our opinion, no strong correlations with the two polymorphisms have been shown, because an individual can be diagnosed with AD over the course of a lifetime and the study of associations is performed precisely on the diagnosis without regard to age. In addition, our subjects received cognitive training, which helped to reduce the progression of cognitive impairment and influenced score changes more than genotype. This suggests that, even though the *APOE* genotype is associated with dementia, with adequate treatment and cognitive training, the onset of severe cognitive effects can be delayed, improving quality of life. Thus, a subject’s genotype can be used to identify individual risk, in order to prescribe preventive cognitive training even before the onset of disease symptoms. In this case, the progression of MCI and the transition to dementia can be delayed almost as much as in people without the “bad” polymorphism.

Based on repeated measures ANOVA, we did not find a significant effect of the *APOE* genotype. In contrast, the general regression model revealed a significant contribution of this polymorphism. At first glance, these results appear conflicting, but these contradictions are eliminated if we understand the meaning of the results obtained. The small number of mutant allele carriers limits the statistical power of the study when we compare several groups. In contrast, regression analysis treats all participants as a single group, which increases the statistical power of the estimates. Our initial regression model estimated the variability of changes in MMSE total score and accounted for group, age and controlled for background MMSE total score. This model explains 25% of the variability in MMSE score changes. Adding *APOE* genotype variable into this initial model adds an additional 3% to the explanatory ability of the model. Thus, we found that *APOE-ε4* polymorphism represents a significant and independent contributor to the 1-year change of total MMSE score in the older subjects.

The limitations of our study were that we did not assess the coronavirus infection in mid- and late 2021, while subsequent infection could also have affected cognitive function. However, it should be noted that the comparison group was recruited from those who presented to the outpatient clinic for vaccination; accordingly, most of the study subjects were vaccinated after one point of follow-up. It was already difficult to distinguish between the presence of antibodies due to the disease and due to vaccination at that time.

## 5. Conclusions

The main findings of our study were:Cognitive training is a positive factor in reducing cognitive impairment.COVID-19 infection had no effect on cognitive function during the 1-year follow-up in older adults.Older adults with *APOE-ε4* genotype showed no positive effect of cognitive training.

Cognitive training is a protective factor against cognitive decline that retains its importance in COVID-19 survivors. Future directions should focus on a more thorough analysis of cognitive dysfunction after COVID-19 in different groups of psychiatric subjects.

The findings are important for predictive models within the positive gerontology paradigm. Similar models contribute to the development of modern methods of specialized prevention of cognitive decline. Attention to the factors discussed in our article, but not limited to them, are useful for a personalized approach to maintaining the cognitive health of older adults.

## Figures and Tables

**Figure 1 diagnostics-12-02312-f001:**
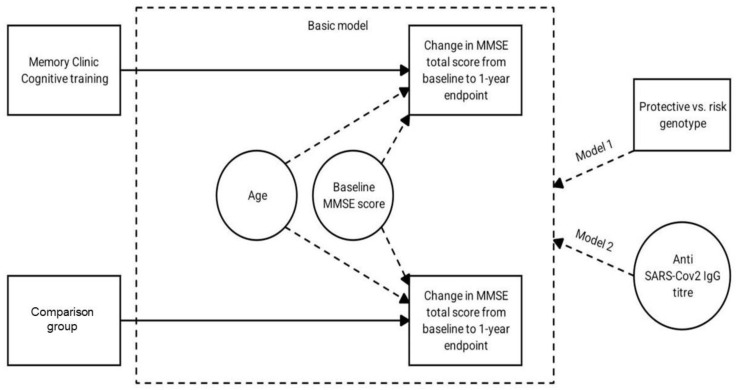
Graphical scheme of research.

**Figure 2 diagnostics-12-02312-f002:**
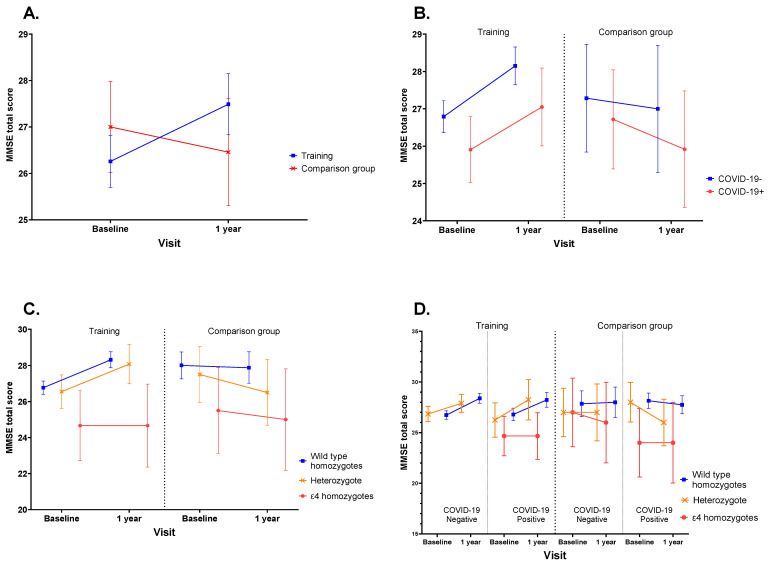
Marginal mean MMSE total score changes for repeated-measures ANOVA for (**A**) time × group interaction, (**B**) time × group × anti-COVID-19 IgG status, (**C**) time × group interaction, (**B**) time × group × *APOE ε4* genotype, (**D**) time × group × anti-COVID-19 IgG status × *APOE ε4* genotype. Note: *p*-levels are provided for within-group comparisons.

**Table 1 diagnostics-12-02312-t001:** General characteristics of study population. Higher education—5 years of education in university.

		Cognitive Training (*n* = 125)	Comparison Group (*n* = 34)	Total (*n* = 159)	Test Statistic
Age, mean (S.D.)		75.0 (5.8)	70.4 (4.6)	74.0 (5.9)	F(1157) = 18.4, *p* < 0.0011
Gender, *n* (%)	Female	114 (91.2)	26 (76.5)	140 (88.1)	$\chi_{1}^{2}$ = 5.51, *p* = 0.022
	Male	11 (8.8)	8 (23.5)	19 (11.9)	
Higher education, *n* (%)	No	43 (34.7)	8 (25.8)	51 (32.9)	$\chi_{1}^{2}$ = 0.88, *p* = 0.352
	Yes	81 (65.3)	23 (74.2)	104 (67.1)	
MMSE baseline total score, mean (S.D.)		26.7 (1.7)	27.9 (1.9)	27.0 (1.8)	F(1157) = 11.78, *p* < 0.0011
MMSE total score change from baseline, mean (S.D.)		1.5 (1.9)	−0.4 (1.5)	1.1 (2.0)	F(1157) = 28.34, *p* < 0.0011
HADS anxiety score, mean (S.D.)		7.0 (3.4)	5.9 (4.0)	6.75 (3.6)	F(1157) = 2.44, *p* = 0.121
HADS depression score, mean (S.D.)		6.3 (3.2)	5.5 (3.2)	6.2 (3.2)	F(1157) = 1.97, *p* = 0.162

**Table 2 diagnostics-12-02312-t002:** Distribution of *APOE* genotype and anti-COVID-19 IgG in study population.

		Group		Total (*n* = 159)	Test Statistics
		Cognitive Training (*n* = 125)	Comparison Group (*n* = 34)		
*APOE ε4*, *n* (%)		24 (19.20%)	5 (14.71%)	29 (18.24%)	Χ^2^ = 1.33, df = 2, *p* = 0.55
	*ε4* homozygotes	3 (2.40%)	2 (5.88%)	5 (3.14%)	
	Wild type homozygotes	98 (78.40%)	27 (79.41%)	125 (78.62%)	
*APOE ε2*, *n* (%)	Heterozygote	18 (14.40%)	7 (20.59%)	25 (15.72%)	Χ^2^ = 0.77, df = 1, *p* = 0.38
	Wild type homozygotes	107 (85.60%)	27 (79.41%)	134 (84.28%)	
IgG to COVID-19, *n* (%)	Positive	37 (29.6%)	24 (70.59%)	61 (38.36%)	Χ^2^ = 18.99, df = 1, *p* < 0.001
	Negative	88 (70.4%)	10 (29.41%)	98 (61.64%)	
IgG to COVID-19, positivity rate, mean (S.D).		2.54 (4.41)	5.76 (4.68)	3.23 (4.65)	F = 13.93, *p* < 0.001

**Table 3 diagnostics-12-02312-t003:** Results of repeated-measures ANOVA of MMSE total score changes from baseline to 1 year by treatment, *APOE-ε4* genotype and anti-COVID-19 IgG factors.

Source	Type III Sum of Squares	df	Mean Square	F	*p*
Time	0.653	1	0.653	0.384	0.536
Time × Group	7.272	1	7.272	4.278	0.040
Time × IgG against COVID-19	0.010	1	0.010	0.006	0.939
Time × *APOE-ε4*	3.180	2	1.590	0.935	0.395
Time × Group × IgG against COVID-19	3.877	1	3.877	2.281	0.133
Time × Group × *APOE-ε4*	3.387	2	1.693	0.996	0.372
Time × IgG against COVID-19 × *APOE-ε4*	1.144	2	0.572	0.336	0.715
Time × Group × IgG against COVID-19 × *APOE-ε4*	2.527	1	2.527	1.487	0.225
Error (Time)	251.597	148	1.700		

**Table 4 diagnostics-12-02312-t004:** Effect of group on change of MMSE total score after adjustment on background HADS depression subscale score.

	DF	Sum of Squares	Mean Square	F-Ratio	*p*-Value	η_p_^2^
Model	2	76.17	38.08	11.06	<0.001	
Baseline Depression HADS score	1	2.44	2.44	0.71	0.401	0.005
Group	1	75.76	75.76	22.00	<0.001	0.14
Error	137	471.72	3.44			
Total(Adjusted)	139	547.89	3.94			

**Table 5 diagnostics-12-02312-t005:** Regression estimates of initial model of prediction of changes MMSE total score based on group factor and MMSE background total score and age.

Effect	B	Standard Error	t	*p*	95% CI	
					Lower	Upper
(Intercept)	13.59	3.31	4.10	<0.001	7.05	20.14
Age	−0.03	0.03	−1.13	0.26	−0.08	0.02
Baseline total MMSE score	−0.37	0.08	−4.46	<0.001	−0.54	−0.21
Group (without training)	−1.59	0.36	−4.43	<0.001	−2.30	−0.88

**Table 6 diagnostics-12-02312-t006:** Effect of adding *APOE* genotype and anti-SARS-CoV-2 IgG on the initial regression model of changes of MMSE total scores.

Model	R^2^	R^2^ Change	df1	df2	*p*
H_0_ (Group; Age; MMSE total score)	0.25	0.25	3	155	<0.001
Adding *ε4* allele of *APOE* to H_0_	0.28	0.04	2	153	0.02
Adding *ε2* allele of *APOE* to H_0_	0.26	0.01	1	154	0.24
Adding IgG against SARS-CoV-2 to H_0_	0.25	0.00	1	154	0.55

Note. H_0_—Null model includes Group, Age, background MMSE total score (Table 3).

## Data Availability

Data available on request.

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
