# Peer review of "Positive Effect of Cognitive Training in Older Adults with Different APOE Genotypes and COVID-19 History: A 1-Year Follow-Up Cohort Study"

_diagnostics, 2022, doi:10.3390/diagnostics12102312_

Round 1

Reviewer 1 Report

In this observational study, Zorkina Y, et al. evaluated the effects of coronavirus infection and APOE genotypes on cognitive changes in mild cognitive patients followed up for one year. The topic seems to be novel and relevant for the field, considering the current context of the COVID-19 pandemics and the preliminary studies suggesting a potential association between APOE genotypes and disease severity. Besides of the evidence suggesting an increased risk to develop dementia for ApoE4 genotype carriers. Nevertheless, the gap in knowledge and objectives of the study are not well established by the authors and there are major flaws in the study design that the authors should address and clarify for a better understanding and interpretation of the results and conclusions.

Therefore, I have some comments for the authors:

I.               Abstract:

1.     Background: Please provide a brief statement on the problematic connecting COVID-19 infection and ApoE genotypes in the MCI setting. Besides, please provide the objective/hypothesis in one clear statement.

2.     Methods: Please define the study design, similar to the study population and the analysis plan. Further defining COVID-19 infection as modifiable is confusing, could be better to define it as acquired risk factor compared to APOE which is innate. Also, could you please state what do you mean by “observed in dynamics”?

3.     Results: This section might be clearer if objective and methods well defined and the included population is not defined.

4.     Conclusions: Please provide a conclusive statement based on the results and not suggestions.

II.             Introduction:

1.     I have the impression that the Information is there, but in disorder, I will suggest using the following order for the evidence:

a.     Start with a brief definition of MCI, without mentioning subtypes.

b.     Then epidemiological information regarding MCI, its prevalence to show its importance and which age group are affected the most. Also, the progression rate to different dementia subtypes.

c.     Potential well known risk factors, mention the APOE and its relationship with dementia and MCI progression.

d.     State the importance of identifying risk factors

e.     Mention the COVID-19 pandemic and how it relates to MCI progression to dementia and the APOE gene subtypes.

f.      Stated the gap in knowledge, propose your hypothesis and briefly explain the plan to solve this gap.

2.     Please clearly stablish the gap in knowledge (important your study), objectives and hypothesis, which are not clear in the manuscript.

3.     Some references seem to be lacking, please add them correspondingly:

a.     Line 66-68: “Since AD is an incurable disease, early diagnosis and identification of risk factors 66 play a major role in therapy. Much attention is given to identifying modifiable and non-67 modifiable risk factors to prevent or inhibit the progression of MCI into dementia.”

b.     Line 69 to 70: According to the disease-oriented model, MCI is a pre-dementia (early) stage in Alz-69 heimer's disease, macro- and microangiopathies of various etiologies.

c.     Line 71: For rehabilitation 70 in dementia, cognitive training (cognitive stimulation therapy, CST, and others) is used.

d.     Line 70 to 80

e.     Line 81 -82

4.     To provide a better support for the reason conduction this study, please add these references in your manuscript:

a.     For MCI and APOE relationship:

Ma Chao et al. Disrupted Brain Structural Connectivity: Pathological Interactions Between Genetic APOEe4 Status and Develovep MCI condition. Mol Neurobiol. 2017 ;54(9):6999-7007. doi: 10.1007/s12035-016-0224-5. Epub 2016 Oct 26.

b.    For COVID and APOE relationship:

Kurki SN, et al. APOE e4 associates with increased risk of severe COVID-19, cerebral microhaemorrhages and post-COVID mental fatigue: a Finnish biobank, autopsy and clinical study.

Espinosa-Salinas I, et al. Potential protective effect against SARS-CoV-2 infection by APOE rs7412 polymorphism. Sci Rep 12, 7247 (2022). https://doi.org/10.1038/s41598-022-10923-4.

Khuo CL, et al. APOE e4 genotype predicts severe COVID-19 in the UK biobank community cohort. J Gerontol A Biol Sci Med Sci. 2020 Nov; 75(11): 2231–2232.

Manzo C, et al. Could COVID-19 anosmia and olfactory dysfunction trigger an increased risk of future dementia in patients with ApoE4?. Med Hypotheses. 2021 Feb; 147: 110479.  (For hypothesis making)

Miners S, et al. Cognitive impact of COVID-19: looking beyond the short term. Alzheimers Res Ther. 2020; 12: 170. 

5.     Not sure about the link between the current study and the concept of positive gerontology. Try make more clear the link. Maybe this could be reduce to just a few lines.

6.     Not sure if  covid-19 should be stated as a modifiable risk factor, since it can be prevented but not modified once obtained. I would be better to define it  just as an exposure since we are not sure if it is actually a risk factor or not. That is the purpose of your study.

7.     Please paraphrase Line 112.

8.     Try to focus more on MCI progression to dementia in general and not just in Alzheimer´s disease, although it is the most common type of dementia.  The issue is that your study is focus on MCI patients and their cognitive deterioration as an outcome and not in their progression to Alzheimer disease per se.

III.            Methods and Results:

1.     Please could you describe better the study design, it seems to be a cohort study. Are you looking for the risk of cognitive deterioration in MCI patients due to COVID-19 and APOE genotypes exposure? If so, please better explain the purpose of having a control group without cognitive therapy.

2.     Based on the above the population is not well defined, are all included subjects with MCI? Or is the control group composed of healthy subjects?

3.     Please could you establish better the first point of observation, and when was the last subject enrolled.

4.     In line 130-131 it is mentioned a first and last follow up, is the first follow up the same as the first observation point? If not, how many follow up periods did the subjects have?

5.     In line 131, please paraphrase this sentence to express that the cognitive tests were performed by the MMSE and besides that the clinical symptomatology of COVID-19 was assessed.

6.     It is great to explain in full details the cognitive rehabilitation patients were receiving however, it is not well stablished the relation of this for the purpose of the study. Are you assessing a potential protective effect of cognitive therapy?

7.     In the statistical analysis, it would be interesting if you could assess the interaction between COVID-19 status and APOE genotype looking for a potential modifying effect of the infection towards the progression of cognitive deterioration.

8. Please provide as much as possible the estimates (ex. means) for directionality and 95% confident intervals.

Author Response

Dear Reviewer 1,

Thank you for taking the time and effort necessary to review our manuscript. We are grateful for such a comprehensive and detailed analysis of our manuscript and for the helpful recommendations. We believe that your suggestions and questions have made a significant contribution to improving our work, and we would like to express our gratitude for this important contribution. We have attempted to incorporate all of the comments and have significantly changed the text of the manuscript. We would like to present our corrections in more understandable way and have highlighted in yellow our addition. We hope that such form of answer will be appropriate.

We have completely rewritten the abstract and hope that in this form it will more accurately represent the results of the study. Also, we fully agree with your comments about the introduction, we have adjusted the introduction according to your comments and supplemented the list of references with the articles you recommended. We also rewrote the materials and methods, included a study design in the results, and now the study design is presented in a clearer way. Additionally we created a study design figure to better understand the design of the study and our main concepts and assumptions.

Additionally, we would like to discuss some selected comments: 

  1. In line 131, please paraphrase this sentence to express that the cognitive tests were performed by the MMSE and besides that the clinical symptomatology of COVID-19 was assessed.

Response: We are grateful to Reviewer 1 for pointing out this deficiency to us. We have adjusted this sentence.

  1.  It is great to explain in full details the cognitive rehabilitation patients were receiving however, it is not well stablished the relation of this for the purpose of the study. Are you assessing a potential protective effect of cognitive therapy?

Response: Cognitive training is a rehabilitation program carried out in our hospital. An additional aim of our work was to investigate the assumption of a positive effect on cognitive function of the rehabilitation program in the presence of other risk factors. We added information about it in introduction, aim and discussion.

  1.  In the statistical analysis, it would be interesting if you could assess the interaction between COVID-19 status and APOE genotype looking for a potential modifying effect of the infection towards the progression of cognitive deterioration.

   Response: We are grateful to Reviewer 1 for helpful advice that qualitatively improved our study. We constructed a regression model that took all these factors into account, and the result was that COVID-19 was not a significant factor. About APOE we made the following conclusion: «Based on repeated measures ANOVA, we did not find a significant effect of the APOE genotype. In contrast, the general regression model revealed a significant contribution of this polymorphism. At first glance, these results appear conflicting. But these contradictions are eliminated if we understand the meaning of the results obtained. The small number of mutant allele carriers limits the statistical power of the study when we compare several groups. In contrast, regression analysis treats all participants as a single group, which increases the statistical power of the estimates. Our initial regression model estimated the variability of changes in MMSE total score and accounted for group, age, and controlled for background MMSE total score. This model explains 25% of the variability in MMSE score changes. Adding APOE genotype variable into this initial model adds an additional 3% to the explanatory ability of the model. Thus, we found that APOE-ε4 polymorphism represent a significant and independent contributor to the 1-year change of total MMSE score in the elderly subjects. Elder adults with APOE-ε4 genotype showed no positive effect of cognitive training»

  1. Please provide as much as possible the estimates (ex. means) for directionality and 95% confident intervals.

Response: As recommended by Reviewer 1, we have supplemented the manuscript with this information in the Figures.

Reviewer 2 Report

In this original paper, the authors show the results of 1-year study on the effects of cognitive training in pazients with MCI, and the potential correlations with APOE gebotype or COVID-19 infection. The most interesting result is the negative effect of APO e4 on the progression of cognitive decline that can be compensated by an appropriate cognitive training. I have some suggestions for the authors:

1) to improve the introduction, citing some papers concerning the worsening of cognitive performance and neuropsychiatric symptoms during Covod-19 infection or lockdown (es. Cagnin A et al. Behavioral and Psychological Effects of Coronavirus Disease-19 Quarantine in Patients With Dementia. Front Psychiatry. 2020 Sep 9;11:578015. doi: 10.3389/fpsyt.2020.578015. PMID: 33033486; PMCID: PMC7509598).

2) to improve the description of methods: the study design should be clearer and of immediate comprehension for the reader

3) finally, to improve the English language. Now, the reading of text is a bit laborious and does not allow for ready trasmission of the message

Author Response

Dear Reviewer 2,

Thank you for taking the time and effort necessary to review our manuscript. We sincerely appreciate all valuable comments and suggestions, which helped us to improve the quality of our work. We tried to take into account all of your recommendations as much as possible and would like to present our corrections in more understandable way. Additions and corrections are highlighted in yellow; we hope this format will be convenient.

Response to Reviewer 2 Comments

  • to improve the introduction, citing some papers concerning the worsening of cognitive performance and neuropsychiatric symptoms during Covod-19 infection or lockdown (es. Cagnin A et al. Behavioral and Psychological Effects of Coronavirus Disease-19 Quarantine in Patients With Dementia. Front Psychiatry. 2020 Sep 9;11:578015. doi: 10.3389/fpsyt.2020.578015. PMID: 33033486; PMCID: PMC7509598).

Response: We are grateful to Reviewer 2 for his valuable advice and supplemented the introduction with a reference to an article that Reviewer 2 recommended to us. In addition, we completely redid the introduction because another Reviewer had comments on the introduction.

2) to improve the description of methods: the study design should be clearer and of immediate comprehension for the reader

Response: We are grateful to Reviewer 2 for pointing out these problem. We rewrote the materials and methods included a study design in the results, and now the study design is presented in a clearer way. Additionally we created a study design figure to better understand the design of the study and our main concepts and assumptions.

  1. finally, to improve the English language. Now, the reading of text is a bit laborious and does not allow for ready transmission of the message

Response: We are grateful to Reviewer 2 for his attention to this problem. We tried to improve the English language.

Reviewer 3 Report

In the study of Zorkina and colleagues, the authors aimed to investigate the effect of the covid-19 infection and apoe genotype on the cognitive decline in MCI subjects after a one-year follow-up, reporting no changes related to the COVID-19 infection and a significant effect of the APOE ε4 genotype on cognitive functions.

The study is well written, the results can be interesting, and the sample is adequate. However, I have some major concerns regarding the methods and the results, especially the discussion, which neglects some results.

·        My major concern regards the point that seems the main result, which is that subjects who underwent cognitive training had statistically significantly less change in the MMSE score after one-year follow-up than subjects without cognitive training. In fact, the authors compare two groups, the cognitive training, and the control group. However, this result is poorly described both in the results section and mainly in the discussion; even the title does not mention the cognitive training and should be modified to underline this result. In addition, the discussion is largely focused on the effect of COVID-19 on cognitive function but, as stated by the authors, they assessed COVID-19 infection only once, without certainty about possible subsequent infections, thus they cannot state they precisely investigate the effect of COVID-19 on cognitive functions.

·        The second crucial concern regards the described improvement in several cognitive domains, but it is unclear how this improvement was certified. In the Methods, the authors stated that the MMSE and the HADS were used to evaluate the subjects, but how can they confirm an improvement in memory, speech, and executive functions? When describing the rehabilitation program, the authors state that it includes tasks for restoration of visual-spatial gnosis, speech, and memory: did they administer a complete neuropsychological battery to test the described functions.

·        Another concern regards the sample selection strategy. The authors do not specify whether all selected subjects were MCI and which criteria they used to define the MCI condition. In addition, the definition of the control group as “visitors of outpatient clinic…who came to periodic health examinations” is not entirely satisfying. In addition, please use the term “subjects” or “individuals” when referring to MCI, avoiding the term “patients”.

·        Other important critical issues that need to be addressed in the discussion are:

-       Page 7, line 245, the authors refer to a “significant improvement in the scores on eight scales and the MMSE overall score”: what are the eight scales they refer to? How the authors evaluate the following described “components of the cognitive sphere”?

-       The sentence on page 7, line 245, “Therefore, the results of the neuropsychological study…indicate a comprehensive improvement in their cognitive sphere and improved functions of the frontal, parieto-temporal, and deep brain structures” is odd and contains some inaccuracies. Since the authors reported only the MMSE score, the study cannot be considered a neuropsychological study unless they will report the results of the neuropsychological assessment they cite. It is not entirely clear whether the author used a comprehensive battery to evaluate cognitive functions or they used only the MMSE. Even if they used an extensive neuropsychological battery, the anatomical definition of the corresponding deficits - frontal, parieto-temporal, and deep brain structures – is out of context in this study, lacking imaging data.

-       From page 7, line 257, to page 8, line 303, the authors describe in a large (maybe too extensive) section previous results about the influence of COVID-19 on cognitive functions, the relationship between COVID-19, dementia and chronic conditions, the cooccurrence of neuropsychiatric symptoms. However, the study's main results show no association between COVID-19 infection and HADS and MMSE. I would suggest reducing the description of premises in this section since their results rebut the association between infection and cognitive decline and depression

-       The sentence “Thus, a patient's genotype can be used to diagnose and identify a patient at risk, in order to prescribe preventive cognitive training even before the onset of disease symptoms” is inaccurate – apoe cannot be used to “diagnose”; in addition, authors should be less assertive, since the study includes a follow-up of only one year, which is insufficient to evaluate long-term cognitive decline. Please also modify the conclusion, as the apoe is not a diagnostic test.

-       The authors should discuss how the baseline MMSE affects the MMSE decline during the follow-up

Other minor concerns:

ABSTRACT

-       I would suggest the authors use a less assertive conclusive sentence since the genotype, in this case, can be used as supportive information and not as a diagnostic tool

INTRODUCTION

-       I suggest the reference Albert 2011 (DOI 10.1016/j.jalz.2011.03.008) for the sentence “amnestic MCI with specific biomarkers is highly suggestive of AD” page 2, line 44.

-       Please use the abbreviation MCI after the first appearance, for example, in page 2, line 52 “Mild cognitive impairment does not mean lifelong low cognitive function” and on page 2, line 53 “In mild cognitive impairment, daily functioning of the patient is mostly intact”

-       Page 2, line 69: the authors describe the positive gerontology paradigm and the use of cognitive stimulation therapy in a long paragraph starting with the sentence “According to the disease-oriented model, MCI is a pre-dementia (early) stage in Alzheimer's disease”. I would suggest the authors add references for their assumptions and clarify better the cited “two approaches” used in rehabilitation

-       In the description of the effect of covid-19 infection (page 3, line 102) I suggest the authors add as references two crucial reviews in the field (DOI: 10.3390/ijerph19137748 and DOI: 10.1007/s12035-020-02249-x); I also suggest to cite the effect of the pandemic overall in people suffering from neurological conditions, resulting in an increased rate of neuropsychiatric disturbances (DOI: 10.3390/jcm10215169)

-       Page 3, line 104, please cite age as a non-modifiable risk factor

-       Page 3, line 120, please add to the sentence “To date, APOE is the 120 only marker associated with MCI to AD progression” “genetic”

METHODS

-       Please specify the time between the first observation (the end of 2020) and the first follow-up (page 3, line 128)

-       Please better describe the study group and the control group, composed of “visitors of outpatient clinic No 121 (Moscow) aged over 65 years who came to periodic health examinations” (page 3, line 128). Where subject MCI, cognitively unimpaired, or SCD? It is important to state whether all included subjects were MCI and which criteria were used for diagnosis

-       Page 4, line 156, please refer to MCI as amnestic single or multiple domains

-       I would suggest the authors describe the neurocognitive program as a separate paragraph from “subjects” paragraph

-       Was the background MMSE score the “baseline” – first visit MMSE (page 4, line 192)? Please specify

RESULTS

-       Please use "baseline" to define "background" MMSE score (page 6, line 226)

TABLES

-       TABLE 1: can the authors report the educational level in years? Are the MMSE scores adjusted for age and educational level or raw score? Please specify. In addition, the MOCA score is reported, but the test has not been described previously in the methods. Please add a description. It would be helpful to detail also the HADS score in table 1

-       TABLE 2: please define the used abbreviations in both tables

FIGURE

Please provide a more detailed legend for the figure, defining the used abbreviation; also describe statistical significance, both in the figure and in the legend 

Author Response

Dear Reviewer 3,

We very much appreciate your taking the time and effort to review our manuscript. We are grateful for the comprehensive and detailed review of our manuscript and for the helpful suggestions that have contributed significantly to improving the text of the manuscript. We have endeavored to incorporate all of the comments and have modified the text substantially. In the process of revising the manuscript text, we have significantly corrected the abstract, introduction, materials, and methods, and additionally we have added a Figure that demonstrates the research design. We wanted to present our corrections in a clearer form and have highlighted our addition in yellow. We hope that this form of response will be appropriate.

Response to Reviewer 3 Comments

       My major concern regards the point that seems the main result, which is that subjects who underwent cognitive training had statistically significantly less change in the MMSE score after one-year follow-up than subjects without cognitive training. In fact, the authors compare two groups, the cognitive training, and the control group. However, this result is poorly described both in the results section and mainly in the discussion; even the title does not mention the cognitive training and should be modified to underline this result.

Response: We are grateful to Reviewer 3 for pointing out this problem. We agree that cognitive training was also one of the goals of our study. We added it to the title and supplemented the objective, discussion, and introduction, as recommended by Reviewer 3. We also expanded on the results and created a study design figure to better understand the design of the study and our main concepts and assumptions.

In addition, the discussion is largely focused on the effect of COVID-19 on cognitive function but, as stated by the authors, they assessed COVID-19 infection only once, without certainty about possible subsequent infections, thus they cannot state they precisely investigate the effect of COVID-19 on cognitive functions.

Response: We fully agree with this statement of Reviewer 3, so we added a limitation to the primary version of the manuscrip. It should also be noted that the patients had been vaccinated, so the chance of them becoming infected was small.

«The limitations of our study were that we did not assess the infection with the coronavirus in mid- and late 2021, while subsequent infection could also have affected cognitive function. However, it should be noted that the control group was recruited from those who presented to the outpatient clinic for vaccination; accordingly, most of the study subjects were vaccinated after 1 point of follow-up. It was already difficult to distinguish between the presence of antibodies due to the disease and due to the vaccination at that time»

      The second crucial concern regards the described improvement in several cognitive domains, but it is unclear how this improvement was certified. In the Methods, the authors stated that the MMSE and the HADS were used to evaluate the subjects, but how can they confirm an improvement in memory, speech, and executive functions? When describing the rehabilitation program, the authors state that it includes tasks for restoration of visual-spatial gnosis, speech, and memory: did they administer a complete neuropsychological battery to test the described functions.

Response: We are grateful to Reviewer 3 for his attention to this problem. In this study, we performed only MMSE and the HADS on subjects. The cognitive training program has been running in our hospital for more than 5 years and the evidence for improvement in the functions listed in the materials and methods has come from our previous studies. We have added a reference to our literature on this. That is why we wrote about it in the materials and methods, and not in the results.

Another concern regards the sample selection strategy. The authors do not specify whether all selected subjects were MCI and which criteria they used to define the MCI condition. In addition, the definition of the control group as “visitors of outpatient clinic…who came to periodic health examinations” is not entirely satisfying.

Response: We are grateful to Reviewer 3 for pointing out this shortcoming. We have corrected the materials and methods that were related to the description of the groups.

In addition, please use the term “subjects” or “individuals” when referring to MCI, avoiding the term “patients”.

Response: We have made a correction as recommended by Reviewer 3.

Other important critical issues that need to be addressed in the discussion are:

-       Page 7, line 245, the authors refer to a “significant improvement in the scores on eight scales and the MMSE overall score”: what are the eight scales they refer to? How the authors evaluate the following described “components of the cognitive sphere”?

-       The sentence on page 7, line 245, “Therefore, the results of the neuropsychological study…indicate a comprehensive improvement in their cognitive sphere and improved functions of the frontal, parieto-temporal, and deep brain structures” is odd and contains some inaccuracies. Since the authors reported only the MMSE score, the study cannot be considered a neuropsychological study unless they will report the results of the neuropsychological assessment they cite. It is not entirely clear whether the author used a comprehensive battery to evaluate cognitive functions or they used only the MMSE. Even if they used an extensive neuropsychological battery, the anatomical definition of the corresponding deficits - frontal, parieto-temporal, and deep brain structures – is out of context in this study, lacking imaging data.

Response: We realized that this information was unnecessary in the manuscript, so we decided to exclude it. We are grateful to Reviewer 3 for these essential questions, which helped us significantly improve the quality of the study by removing unnecessary information.

From page 7, line 257, to page 8, line 303, the authors describe in a large (maybe too extensive) section previous results about the influence of COVID-19 on cognitive functions, the relationship between COVID-19, dementia and chronic conditions, the cooccurrence of neuropsychiatric symptoms. However, the study's main results show no association between COVID-19 infection and HADS and MMSE. I would suggest reducing the description of premises in this section since their results rebut the association between infection and cognitive decline and depression

Response: We are grateful to Reviewer 3 for his valuable advice, but we think it is important to demonstrate this association. This section is written to show how many studies have shown a positive result. Indeed, our study did not show this despite the premise. However, we think it is important for the reader to know this.

-       The sentence “Thus, a patient's genotype can be used to diagnose and identify a patient at risk, in order to prescribe preventive cognitive training even before the onset of disease symptoms” is inaccurate – apoe cannot be used to “diagnose”;

Response: We are grateful to Reviewer 3 for pointing out our mistake. We have made the appropriate correction.

 in addition, authors should be less assertive, since the study includes a follow-up of only one year, which is insufficient to evaluate long-term cognitive decline. Please also modify the conclusion, as the apoe is not a diagnostic test.

Response: We have made corrections according to the recommendations of Reviewer 3, and corrected for softer sounding - instead of diagnosis - the criterion of individual risk: «the marker presence can only be used as a criterion of individual risk of disease for as early as possible cognitive therapy»

We also added the phrase "during the 1-year follow-up: «COVID-19 infection had no effect on cognitive function during the 1-year follow-up in elder adults. »

The authors should discuss how the baseline MMSE affects the MMSE decline during the follow-up

Response: We are thankful to Reviewer 3 for his attention to this problem. We considered the effect of baseline MMSE in the regression model. However, we did not conduct such a study separately, because it was not the task of this study.

Other minor concerns:

ABSTRACT

-       I would suggest the authors use a less assertive conclusive sentence since the genotype, in this case, can be used as supportive information and not as a diagnostic tool

Response: We are grateful to Reviewer 3 for his attention to this problem. We have completely corrected the abstract and hope that the new version will better represent the results of the study.

INTRODUCTION

-       I suggest the reference Albert 2011 (DOI 10.1016/j.jalz.2011.03.008) for the sentence “amnestic MCI with specific biomarkers is highly suggestive of AD” page 2, line 44.

Response: We have supplemented the manuscript with the reference suggested by Reviewer 3.

-       Please use the abbreviation MCI after the first appearance, for example, in page 2, line 52 “Mild cognitive impairment does not mean lifelong low cognitive function” and on page 2, line 53 “In mild cognitive impairment, daily functioning of the patient is mostly intact”

Response: We have made a correction as recommended by Reviewer 3.

-       Page 2, line 69: the authors describe the positive gerontology paradigm and the use of cognitive stimulation therapy in a long paragraph starting with the sentence “According to the disease-oriented model, MCI is a pre-dementia (early) stage in Alzheimer's disease”. I would suggest the authors add references for their assumptions and clarify better the cited “two approaches” used in rehabilitation

Response: We have made corrections as recommended by Reviewer 3, and we have revised this paragraph: «According to the new paradigm of positive gerontology, the emphasis is not on disease as an obligate link in aging, but on the ability to use mental resources in the third and fourth age. The development of modern approaches to rehabilitation of cognitive deficits uses predictive models with multimodal markers, which is important for medical professionals - for diagnosis, identification of therapeutic and rehabilitation targets, as well as non-medical specialists who work with the problem of cognitive competence decline at the stage of late ontogenesis. Genetic and environmental factors are being studied both in connection with the development of neurodegenerative and cerebrovascular diseases and in connection with the re-organization of cognitive resources. The coronavirus pandemic is a new serious factor in cognitive decline, the role of which requires a long study»

-       In the description of the effect of covid-19 infection (page 3, line 102) I suggest the authors add as references two crucial reviews in the field (DOI: 10.3390/ijerph19137748 and DOI: 10.1007/s12035-020-02249-x); I also suggest to cite the effect of the pandemic overall in people suffering from neurological conditions, resulting in an increased rate of neuropsychiatric disturbances (DOI: 10.3390/jcm10215169)

Response: We have supplemented the text with the references that Reviewer 3 suggested.

-       Page 3, line 104, please cite age as a non-modifiable risk factor

Response: We have made corrections as recommended by Reviewer 3.

-       Page 3, line 120, please add to the sentence “To date, APOE is the 120 only marker associated with MCI to AD progression” “genetic”

Response: Since we have completely redone the introduction according to Reviewer 1's comments, there is another sentence about it in the new version, it goes like this: «То date ε-polymorphism of the apolipoprotein E (APOE) gene remains the strongest genetic factor affecting the risk of AD after numerous large-scale genomic association studies and genomic association meta-analysis»

METHODS

-       Please specify the time between the first observation (the end of 2020) and the first follow-up (page 3, line 128)

Response: We are grateful to Reviewer 3 for pointing out this deficiency to us. This information should be presented in the manuscript, so we have added it.

-       Please better describe the study group and the control group, composed of “visitors of outpatient clinic No 121 (Moscow) aged over 65 years who came to periodic health examinations” (page 3, line 128). Where subject MCI, cognitively unimpaired, or SCD? It is important to state whether all included subjects were MCI and which criteria were used for diagnosis

Response: We are thankful to Reviewer 3 for bringing this problem to our attention. We have corrected the description of the groups, and we hope that this way it will be more understandable. We removed the word «control group» because it does not quite fit the concept of a control group and replaced it with «comparison group».

-       Page 4, line 156, please refer to MCI as amnestic single or multiple domains

Response: To improve reading comprehension of the manuscript, we have replaced this with “cognitive decline”. We hope that these corrections will be appropriate.

-       I would suggest the authors describe the neurocognitive program as a separate paragraph from “subjects” paragraph

Response: We have made a correction as recommended by Reviewer 3.

-       Was the background MMSE score the “baseline” – first visit MMSE (page 4, line 192)? Please specify

Response: Yes, first visit MMSE was the baseline. We made a correction accordingly.

RESULTS

-       Please use "baseline" to define "background" MMSE score (page 6, line 226)

Response: We have made a correction as recommended by Reviewer 3.

TABLES

-       TABLE 1: can the authors report the educational level in years? Are the MMSE scores adjusted for age and educational level or raw score? Please specify.

Response: We are grateful to Reviewer 3 for his attention to this problem. We know that traditionally the indicator for years of education is used. But we decided to move away from this, and introduce the indicator of the presence or absence of higher education. Because in Russia in this period of time studied in school for 10 years, and then 5 years in higher education. Having higher education corresponds to 15 years of study. The MMSE scale in Table 1 has not been adjusted for age and education level. But age was included in the regression model.

In addition, the MOCA score is reported, but the test has not been described previously in the methods. Please add a description. It would be helpful to detail also the HADS score in table 1

Response: We are grateful to Reviewer 3 for pointing out these inconsistencies in the text of the manuscript. We removed the MOCA score from this table. It should not be in the text. We also described the HADS score in Table 1.

-       TABLE 2: please define the used abbreviations in both tables

Response: We have made a correction as recommended by Reviewer 3.

FIGURE

Please provide a more detailed legend for the figure, defining the used abbreviation; also describe statistical significance, both in the figure and in the legend 

Response: We decided to change the Figure considerably; it seems to us that in this form it will be more informative. We hope that this version of the Figure will be acceptable.

Round 2

Reviewer 1 Report

In this observational study, Zorkina et al. evaluated the association of COVID-19 infection and APOE genotype with cognitive decline in elderly adults as well as the impact of cognitive training. The topic seems to be novel and relevant to the field, considering the current context of the COVID-19. Comparing the previous manuscript, I must congratulate the authors for the significant improvements they have done in the current one. The research study is presented in much clear way, the evidence in the introduction is presented appropriately making possible to identify gap in research besides of the main objectives. Furthermore, the authors have improved the explanation of the methods allowing the reader to understand better the results and discussion.

I just have some minor suggestions:

1)    Methods:

-       Suggestion 1: In Lines 169-171, the authors mentioned:

We did not distinguish the group with normal cognitive functions from the subjects of the comparison group, based on the purpose of our study and because the statistical methods we used allow us to solve such problems.

I think there might be a potential typo since the normal cognitive function group and the comparison one seems to be the same. I think you might want to say the abnormal cognitive functions group or group with subjective cognitive decline. If there was no typo, I think this phrase might cause confusion and I recommend rephrasing it to make it more clear.

-       Suggestion 2:  I recommend mentioning figure 1 (currently in results) in the methods section (statistical analysis part).

2)    Results:

-Suggestion 1: In table 3, there is a variable coded as “visit”, I think you are refering here to “time”. Therefore, it would be better if you change it for “time” to make it much easier to understand.

-Suggestion 2: I think it will sound better if you change the word mathematically in line 249 and line 297 for arithmetically.

Author Response

Dear Reviewer 1,

We thank you for your positive comments on our manuscript! We sincerely appreciate your help in correcting the presentation of our study and are grateful for the time and effort you put into it.

We are thankful for additional suggestions on how to improve our manuscript. We have made several corrections according to your comments.

1)    Methods:

-       Suggestion 1: In Lines 169-171, the authors mentioned:

“We did not distinguish the group with normal cognitive functions from the subjects of the comparison group, based on the purpose of our study and because the statistical methods we used allow us to solve such problems.”

I think there might be a potential typo since the normal cognitive function group and the comparison one seems to be the same. I think you might want to say the abnormal cognitive functions group or group with subjective cognitive decline. If there was no typo, I think this phrase might cause confusion and I recommend rephrasing it to make it more clear.

Response: We fully agree that this phrase is not correct, so we decided to remove it so as not to deceive the readers of the manuscript. 

-       Suggestion 2:  I recommend mentioning figure 1 (currently in results) in the methods section (statistical analysis part).

Response: We have added a mention of Figure 1 to the methods according to your recommendation.

2)    Results:

-Suggestion 1: In table 3, there is a variable coded as “visit”, I think you are refering here to “time”. Therefore, it would be better if you change it for “time” to make it much easier to understand.

Response: We have made corrections according to your recommendations.

-Suggestion 2: I think it will sound better if you change the word mathematically in line 249 and line 297 for arithmetically.

Response: We have made corrections according to your recommendations.

Reviewer 3 Report

The authors replied to all my points. Howevere, there are still few concerns needing to be addressed. The study population includes individuals with subjective cognitive decline, but the introduction and the discussion is focused on subjects with mild cognitive impairment. Please provide a reference for subjective cognitive decline definition (how where the individuals included? If they were selected based on specific diagnostic criteria for subjective cognitive decline, it should specified in the methods). Please provide a brief description of subjective cognitive decline as a frequent condition in elderly. 

Please provide a definition for "higher education" in the legend of table 1. 

Author Response

We thank you for your positive evaluation of our manuscript. We sincerely thank you for your help in improving our manuscript and for your additional comments. We have made several corrections according to your comments.

The authors replied to all my points. Howevere, there are still few concerns needing to be addressed. The study population includes individuals with subjective cognitive decline, but the introduction and the discussion is focused on subjects with mild cognitive impairment. Please provide a reference for subjective cognitive decline definition (how where the individuals included? If they were selected based on specific diagnostic criteria for subjective cognitive decline, it should specified in the methods). Please provide a brief description of subjective cognitive decline as a frequent condition in elderly. 

Response: We are grateful for your attention to this problem and have added an extra paragraph to the manuscript: «The study included individuals with complaints of forgetfulness, lack of attention and concentration (for example, when talking or reading a book), episodic difficulties in finding their way home, feeling hard to articulate their thoughts, decreased professional and social productivity, impaired motor skills (writing, drawing), problems in everyday life (paying bills, shopping).»

Please provide a definition for "higher education" in the legend of table 1. 

Response: We supplemented the legend of Table 1 with the following description: «Higher education – 5 years of education in university». The problem is that when the patients under study were young, at that time in the USSR all students studied for 5 years in higher educational institutions and received a "Specialist" degree. This was considered higher education, and there was no division into bachelor's and master's degrees. Thus, those who have "Higher education" - they all completed 5 years of study at a university as a specialist.